# Memristor-Based Read/Write Circuit with Stable Continuous Read Operation

Weijun Lu *, Ning Bao, Tangren Zheng, Xiaorui Zhang and Yutong Song

School of Electronic Engineering, Beijing University of Posts and Telecommunications, Beijing 100876, China; baoning@bupt.edu.cn (N.B.); ztr2020140375@bupt.edu.cn (T.Z.); zhangxiaorui2015@bupt.edu.cn (X.Z.); songyutong@bupt.edu.cn (Y.S.)
* Correspondence: luwj@bupt.edu.cn; Tel.: +86-13811879453

**Abstract:** In recent years, computation-intensive applications, such as artificial intelligence, video processing and encryption, have been developing rapidly. On the other hand, the problems of "storage wall" and "power consumption wall" for the traditional storage and computing separated architectures limit the computing performance. The computational circuits and memory cells based on nonvolatile memristors are unified and become a competitive solution to this problem. However, there are various problems that prevent memristor-based circuits from entering practical applications, one of which is the memristor state deviation problem caused by continuous reading. In this paper, we study some circuits studied by predecessors on read/write circuit, compare the experimental results, analyze the reason for the resistance state deviation of memristor, and put forward a new parallel structure of memristor based on opposite polarity. The logic "1" and logic "0" are represented by the positive and negative voltage difference of two memristors with opposite polarity, which can effectively alleviate the problem of the resistance state deviation caused by continuous reading. A reading voltage of 2 V is applied to the four circuits at the same time, and continuous reading is carried out until the output voltage becomes stable. The voltage offset of the optimized circuit when reading logic "0" is reduced to 78 mV, which is significantly smaller than that of other circuits. In addition, when reading logic "1", it has the effect of enhancing the information stored in the memristor.

**Keywords:** memristor; read write circuit; state offset





## 1. Introduction

In recent years, the rapid development of big data and deep learning has greatly increased the demand for computing capability. The computing capability required for neural network training of massive data increase exponentially. On the other hand, hardware development has hit a bottleneck. As a result, it is difficult to meet the demand of various computationally intensive applications. There are two main reasons: first is the stagnation of the development of Moore's law. After entering 7 nm, Moore's law gradually stagnates. As the size decreases, the process cost increases seriously, and there will be intrinsic physical law restrictions when entering the atomic scale [1,2]. Another reason is the memory computing separation architecture of traditional computing. The existing computers are a traditional Von Neumann structure. The storage module of this structure is separated from the computing part. The computing power of the hardware can be solved by increasing the parallelism of the computing unit, while the increase of the computing unit increases the data transmission requirements between the data path and the storage unit. This leads to the problem that the data transmission speed between the two modules is much lower than the calculation speed, that is, the problem of "storage wall" [3,4].

As early as the beginning of the exposure of storage wall and memory access power consumption (1990s), researchers began to find solutions or weakening methods. From the initial multi-level storage architecture [5,6], to near storage computing [7,8], until the

integration of storage and computing, researchers have conducted a lot of work. The first two do not fundamentally solve the problem of storage wall, but only reduce the distance between storage modules and computing modules to alleviate the problem of storage wall as much as possible. The integration of storage and computing is to integrate computing and storage into one module. This method eliminates the step of data transmission and can fundamentally solve the problem of storage wall. However, there was no breakthrough in this scheme until 2008, when HP researchers produced the physical model of memristor for the first time [9,10].

The resistance of the memristor, i.e., memory resistor, will change with the flowing charge, and its resistance will remain unchanged after power supply is stopped. The concept of the memristor was first put forward by professor Shaotang Cai in 1971 [11], and it was considered that the memristor was the fourth basic element besides resistance, capacitance and inductance. The change of the resistance of the memristor is related to the amount of charge flowing through the memristor, that is, when the net charge passing through the memristor is negative, the resistance of the memristor increases, and when the net charge passing through the memristor is positive; therefore, the resistance of the memristor decreases. This nonvolatile nature of the memristor provides the possibility for signal storage and is expected to break the traditional Von Neumann structure and realize the integration of storage and computing, then the problem of "storage wall" of signal transmission caused by the separation of storage and computing [12] could be solved. Since then, a large number of scholars have participated in the research of memory computing integrated system based on memristor.

The logic circuit, read–write circuit, and reset circuit are the basis of constructing the memory computing integrated circuit based on memristor. In 2010, researchers at HP Labs realized the implicit state logic based on memristor and proved that this calculation was logically complete. This shows that the data in the memristor based storage structure can be directly processed through state logic calculation, so as to realize the organic integration of operation and storage [13–17]. Among them, the basic function of the read–write circuit based on the memristor is to ensure that the resistance state of the memristor can be changed during the write operation, and the information represented by the current memristor resistance state can be accurately read during the read operation, and the original resistance state of the memristor cannot be changed. However, in practice, if the memristor is read, voltage will be applied at both ends of the memristor, and the resistance of the memristor will change slightly with the flow of electric charge or magnetic flux. Due to the repeat read operation of the memristor, this weak change of the memristor will continue to accumulate, which will produce unexpected resistance drift. Although this drift may have little effect on binary storage or programming, the error cannot be ignored in multi-level or even arbitrary resistance storage or programming.

In order to eliminate this offset, scholars have carried out relevant research and achieved remarkable results [18–34]. They can be divided into two categories: one is completely composed of memristors. Yenpo Ho et al. [18] proposed a structure in which the write circuit is composed of a memristor, and the readout circuit is composed of a memristor and ordinary resistor in series, and added a reload circuit to alleviate the state offset of memristor. I. E. Ebong et al. introduced an adaptive read–write erase method in [19], which can be used to realize a more flexible storage system in the case of low output in the field of nanotechnology. M. Elshamy et al. proposed a new solution in [20], using three terminal memristor to replace two terminal memristor. Later, they proposed a new non-destructive readout circuit method in [21,22], which adopts the structure of two memristors in series, which can enhance the information stored in the memristor when reading logic "1", but cannot solve the problem of state offset when reading logic "0". In 2014, M. Abdullah et al. evaluated the impact of process changes on the electrical performance of TiO2 thin film memristor through modeling analysis and Monte Carlo simulation [23] and improved the writing and reading efficiency of the device by finding the best writing input flux and the best threshold, respectively. S. F. Nafea et al. proposed a memory read–write circuit based

on spintronics memristor in [24]. The proposed read–write circuit can significantly reduce the occupied area.

The other category is composed of memristors and MOS transistors. B.R. Biswas et al. developed a four-memory array design with complete read–write technology based on the memristor MOS hybrid structure in [25]. The write technology based on data erasure proposed in this design replaces the write technology based on feedback reading and reduces the circuit complexity. The circuit proposed by Mohammad Nazmus Sakib et al. [26] consists of a memristor and six MOS transistors. During the read operation, the voltage in the same direction as when writing "0" is introduced, and extensive simulation experiments are carried out on the model transistor using 32 nm prediction technology. The memory cell has good performance and stability in terms of read–write time and switching power consumption. The circuit proposed by Soumitra Pal et al. in [27] consists of two memristors and four MOS transistors. Two memristors are connected in parallel with opposite polarity. One memristor plays the role of storage, the other memristor plays the role of auxiliary, and MOS tube plays the role of switching. The circuit can only alleviate the state offset problem to a certain extent, but it has not been fundamentally solved. It can be seen that the previous research results in the read–write circuit based on memristor are quite remarkable, but these results only add new functions or optimize the performance of the circuit. There is no radical solution to the memristor state offset caused by continuous reading.

The structure of this article is organized as follows. Section 2 briefly explains the motivation and contribution of this paper. Section 3 introduces the principle of the memristor in detail, and analyzes the mechanism and existing problems of the circuit proposed in [18,21,27]. Section 4 introduces the principle and advantages of the circuit proposed in this paper. Section 5 verifies the feasibility of the proposed circuit through experiments. Section 6 summarizes this article.

## 2. Motivation

The basic function of the read–write circuit based on the memristor is to ensure that the resistance state of the memristor can be changed during the write operation, and the information represented by the current memristor resistance state can be accurately read during the read operation, and the original resistance state of the memristor cannot be changed. However, in practice, when reading the memristor, voltage will be applied at both ends of the memristor, which will cause the resistance of the memristor to change slightly with the amount of charge or magnetic flux flowing. Due to the repeat read operation of the memristor, this weak change of the memristor will continue to accumulate, which will produce unexpected resistance drift. Although this drift may have little effect on binary storage or programming, the error cannot be ignored in multi-level or even arbitrary resistance storage or programming.

Over the last few years, many read/write circuits have been proposed to solve the problem of memristor resistance offset caused by continuous reading operations. The circuit of Nonvolatile memristor memory in [18] (hereinafter referred to as NMM) is a classic read/write circuit based on the memristor in this field. The structure of NMM is relatively simple and can realize the functions of writing and reading, the problem of NMM is that the state offset is very serious in the condition of continues reading of logic "0". The author uses a reload circuit to overcome the problem. The addition of reload circuit makes the whole circuit more complex and decreases the performance, so this solution has little practicality. A nondestructive read/write circuit for memristor-based memory arrays in [21] (hereinafter referred to as NRC) is an improvement of NMM. When reading logic "1", the circuit does not cause the problem of memristor state offset. Moreover, it makes the resistance state in the memory memristor more stable. However, the problem of memristor state offset still exists when reading logic "0". The structure of High reliability read–write circuit based on memristor in [27] (hereinafter referred to as HRC) adds an auxiliary memristor. HRC uses MOS transistor and memristor to jointly control the read and write operations. The output of the circuit is represented by the difference between the voltages at both ends of the storage memristor and the auxiliary memristor. This

method only improves the reading accuracy by changing the decoding method of output voltage to a certain extent, and does not fundamentally solve the problem of resistance state offset of memristor.

We propose a memristor-based stable continuous reading circuit (SCRC). The framework of the circuit is composed of two opposite structures in parallel, and each circuit is composed of two memristors with opposite polarity in series. The series connection of two memristors with opposite polarity can effectively alleviate the problem of memristor state shift caused by continuous reading, and this structure helps to restore the memristor to its initial state when the reverse voltage is applied. The parallel connection of two opposite structures is to obtain two voltages with large difference, and the output value of the circuit is the difference between the two voltages. If the difference is positive, the reading result is logic "1"; if the difference is negative, the reading result is logic "0". This method can effectively improve the reading accuracy. Moreover, our circuit has a positive feedback effect when reading logic "1". The experimental results show that after continuous reading, the offset of our structure output voltage is reduced to 78 mv. The impact of this level of offset on the final reading result is almost negligible, and it is significantly lower than the lowest offset (198 mv) of the three circuits in [18,21,27]. In terms of reading delay and power consumption, our circuit is significantly better than those in [18,27]. A detailed principal analysis will be described below.

## 3. Related works

### 3.1. Principle and the Model of Memristor

The resistance of a memristor, also known as memory resistor, is continuously adjustable. The concept of a memristor was first proposed by Shaotang Cai in 1971. He believes that in addition to resistance, capacitance and inductance, there is also a "memristor" representing the relationship between magnetic flux and charge in nature. The effect of this component is that the resistance can change with the current flowing through, and the resistance will remain unchanged after the current stops. Figure 1 shows the physical structure of the memristor (left) and its equivalent circuit model (right). The device is composed of a "metal semiconductor metal" structure. The semiconductor in the middle is composed of doped and undoped regions. The total length of the semiconductor is $D$, the length of the doped region is w, and the boundary between the doped and undoped regions will shift with the passing current. Applying a forward voltage will cause the boundary to shift towards the undoped region, and the resistance of the device will gradually decrease. When the length of the doped region is extended to $D$, the memristor will show a low resistance state ($R_{on}$). After applying a reverse voltage, the boundary will shift towards the doped region, and the resistance of the device will gradually increase. When the length of the undoped region is extended to $D$, the memristor will show a high resistance state ($R_{off}$).

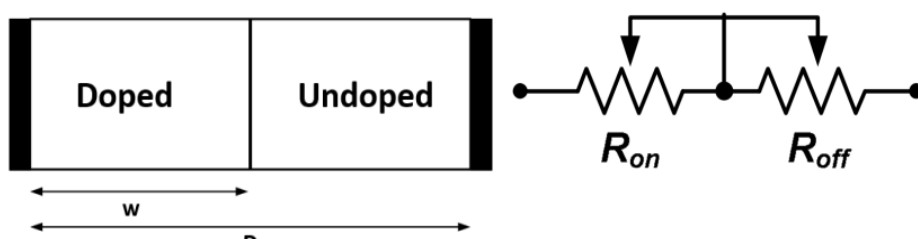

**Figure 1.** Physical structure and equivalent circuit of memristor [18].

Therefore, the resistance value of the memristor can be expressed by Equation (1):

$$R(w) = \left( R_{on} \cdot w/D + R_{off} \cdot (1 - w/D) \right) \tag{1}$$

Figure 2 is the resistance state simulation diagram of the memristor. The mathematical model of the memristor is:

$$M(x) = R_{on}x + R_{off}(1 - x) \tag{2}$$

$$\frac{dx}{dt} = \frac{\mu_v R_{on}}{D^2} i(t) f(x) \tag{3}$$

$$f(x) = 1 - (2x - 1)^{2p} \tag{4}$$

where $\mu_v$ is the migration coefficient, $D$ is the film width of memristor, $f(x)$ is the window function, and Equation (4) is its functional expression. The purpose of the window function is to better fit the nonlinearity of the real situation. The function of the window function is to limit the variable $x$ between 0 and 1. As $x$ gets closer to the boundary value, the value of $f(x)$ becomes smaller. When $x$ is equal to the boundary value 0 or 1, the value of $f(x)$ becomes 0. The control parameter $p$ in the window function controls the linearity of the model. When $p$ increases, the linearity of the model increases. When $p$ is large enough, $f(x) \approx 1$, the nonlinear model changes into a linear model. The simulation parameters of the model are set as $R_{on} = 100\ \Omega$; $R_{off} = 16\ \text{k}\Omega$; Rinit = 11 k$\Omega$; $D = 10$ nm; $\mu_v = 10$ F; $p = 1.0$.

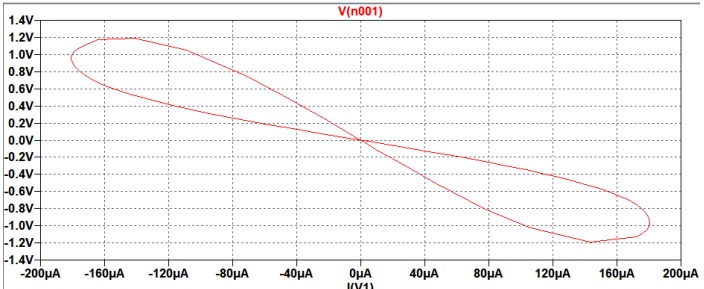

**Figure 2.** Resistance state simulation diagram of memristor.

It can be seen from the Figure 2 that the resistance value of the memristor varies with the change of the applied voltage which leads to two resistance states. When the voltage value reaches the threshold voltage, the memristor switches between the high resistance state and the low resistance state. That is, when the applied forward voltage reaches the threshold voltage, the memristor changes from high resistance state to low resistance state. When the applied reverse voltage reaches the threshold voltage, the memristor changes from low resistance state to high resistance state.

The emergence of memristor makes it be possible to realize the idea of memory computing integration. As an important part, the read–write circuit based on memristor has attracted the interest of a large number of scholars. This paper selects three kinds of classic circuits from the previous research for analysis, and briefly summarizes the advantages and disadvantages of the three circuits in the above introduction. Furthermore, the mechanisms and characteristics of the four circuits will be analyzed in detail below, following that, a new structure is proposed and evaluated with the traditional ones.

### 3.2. Nonvolatile Memristor Memory (NMM)

Nonvolatile memristor memory (NMM) was proposed by Yenpo Ho et al. in [18] in 2011. Figure 3 shows the schematic diagram of the circuit. It can be seen that the structure of the circuit is relatively simple and mainly consists of two parts: the left half is the convert stage, which realizes the writing and reading of signals; the right part is the sense amplifier stage, which is used to amplify the read signal and convert it into digital output signal. Here, we will not introduce the sense amplifier stage too much, but mainly analyze the convert stage.

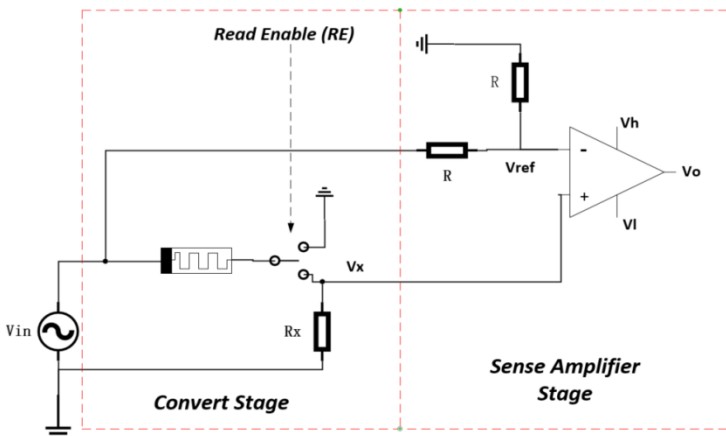

**Figure 3.** Nonvolatile memristor memory (NMM) [18].

The convert stage realizes the writing and reading of signals. As can be seen from Figure 3, the switch is turned to the grounded end to realize the write operation; that is, after the forward voltage is applied, the memristor changes to the low resistance state. At this time, the memristor stores the signal "1". After the reverse voltage is applied, the memristor changes to the high resistance state. At this time, the memristor stores the signal "0", so this method uses the resistance state of the memristor to represent the stored information, that is, the low resistance state ($R_{on}$) represents the logic "1" and the high resistance state ($R_{off}$) represents the logic "0". This writing circuit has no major disadvantages. Therefore, later scholars basically adopted this method or made some improvements, so we will not repeat the writing circuit too much here. The research focus mainly on the readout part. There is a problem which is difficult to solve in the readout circuit. The resistance of the memristor will change after continuous reading, which will affect the information stored in the memristor.

The disadvantage of NMM is that the resistance state deviation of the memristor caused by continuous reading. It can be seen from Figure 3 that when the switch is turned to the end connected with the resistor, the readout operation is realized. At this time, the circuit structure is that the memristor is connected in series with a fixed value resistor, and the voltage between them is output as the read signal. According to reference [18], when logic "1" is stored in the memristor, the memristor is in a low resistance state. According to the voltage division principle, the voltage value on the memristor is approximately zero, so the output voltage Vx is approximately equal to Vin; that is, the output signal "1". When the logic "0" is stored in the memristor, the memristor is in a high resistance state, and the resistance value of the memristor is much greater than the fixed resistance Rx. Therefore, according to the voltage division principle, the voltage value on the memristor is approximately equal to Vin, so the output voltage Vx is approximately equal to zero, that is, the output signal "0". In order to solve the resistance state offset of memristor caused by continuous reading, the read signal Vin is set as two parts: negative pulse and positive pulse, and the duration of the two pulses is the same. However, during the positive pulse period, the internal state w of the memristor increases and the memristor value decreases. In the subsequent positive pulse period, the internal state w of the memristor decreases and the memristor value increases. All this is in the ideal case, the total magnetic flux changes to zero in a read signal cycle, so after the reading, the memristor can return to the initial state. In fact, this is because the memristor decreases in the positive pulse cycle. When the negative pulse period comes, the partial voltage on the memristor will decrease, and the total flux increased on the memristor in the whole positive pulse period will be less than the total flux decreased on the memristor in the negative pulse period. In general, the state variable w of the memristor cannot be restored to the initial state.

### 3.3. Nondestructive Read/Write Circuit for Memristor-Based Memory Arrays(NRC)

Nondestructive read/write circuit for memristor-based memory arrays (NRC) was proposed by Mohamed elshamy et al. in [21] in 2015. Figure 4 is the schematic diagram of the circuit. The circuit is improved on the basis of NMM [18] to solve the problem of resistance state offset of memristor caused by continuous reading. As can be seen from Figure 4, the writing circuit of this circuit is the same as that in NMM. The improvement of this circuit is that the fixed resistance in NMM is replaced by a memristor in a high resistance state. The two memristors are connected in series in the opposite direction, and a diode is added between the two memristors.

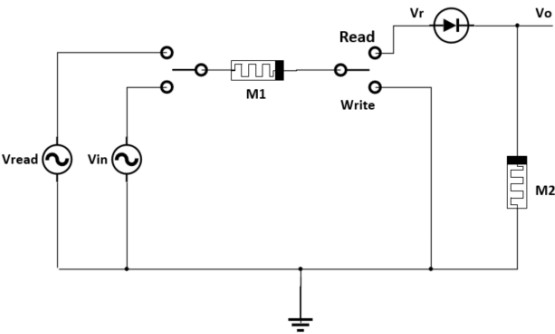

**Figure 4.** Nondestructive read/write circuit for memristor-based memory array (NRC) [21].

The reading signal of this scheme no longer uses the reading signal composed of positive and negative pulse signals with the same period proposed in document [18], but uses a single positive pulse signal. When the data stored in memristor are logic "1", the memristor is in the low resistance state. At this time, the voltage Vr at the front end of the diode is close to Vread. When the diode is turned on, there is a voltage drop Vd, and the voltage drop value is about 0.7 V. Therefore, we must design an appropriate reading signal Vread to ensure that the Vr value is greater than 0.7 V to ensure that there is current flowing through M2. The memristor M2 is used as a load resistance in the circuit to replace the fixed resistance, and the resistance value is $R_{off}$. This can reduce the occupied area of the circuit and reduce the power consumption. By reading the voltage $V_o$ on M2, we can get the state of memristor. Since M2 resistance is fixed at $R_{off}$, we can calculate $V_o$ according to the circuit. When the data stored in the memristor are logic "0", the memristor is in the high resistance state. At this time, the partial voltage of M1 is extremely high which will cause the diode to close and the output voltage $V_o$ is approximately zero.

When reading logic "1", NRC not only does not destroy the information stored in the memristor, but also has the function of rewriting the information. When reading logic "0", the information stored in the memristor will not be affected because the diode is turned off. When reading logic "1", memristor M1 is in the low resistance state and memristor M2 is in the high resistance state. After reading signal Vread is connected, it is equivalent to applying forward voltage to M1 and reverse voltage to M2. M1 shifts to the undoped region and M2 shifts to the doped region. Therefore, the circuit does have the function of rewriting when reading logic "1". When reading logic "0", both memristors M1 and M2 are in a high resistance state, and the value of Vr is less than the voltage drop of the diode by 0.7 V, resulting in the closing of the diode. In fact, at this time, there is still a weak current passing through the memristor, and the memristor will still shift to the undoped region, so the resistance of memristor M1 will still decrease. Therefore, the circuit still does not solve the problem of resistance state offset when reading logic "0".

### 3.4. High Reliability Read–Write Circuit Based on Memristor(HRC)

The structure of High reliability read–write circuit based on memristor (HRC) was proposed by soumitra pal et al. in [27] in 2019. Figure 5 shows the schematic diagram of the circuit. As can be seen from the Figure 5, this circuit adopts a completely different connection mode from the circuits presented in Sections 3.2 and 3.3. Two memristors with

opposite polarity are connected in parallel. Memristor M1 is used as a storage device and memristor M2 is used as an auxiliary device. Several transistors play the role of switching control. The data information stored in the memristor M1 is determined by comparing the difference between nodes Y2 and Y3.

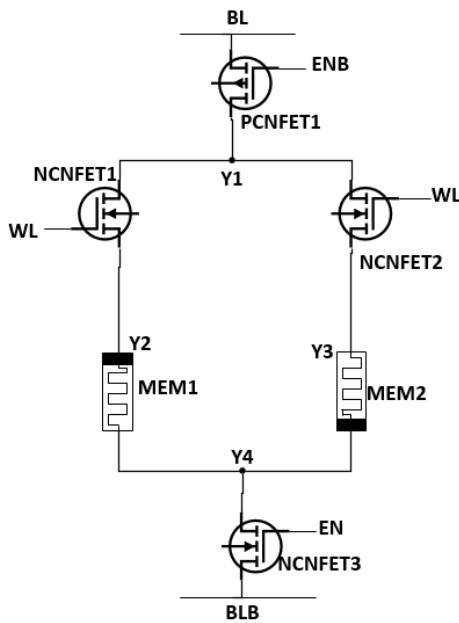

**Figure 5.** High reliability read–write circuit based on memristor(HRC) [27].

During the write operation, WL and EN are set as the power supply voltage VDD, while ENB is set as GND to turn on transistors NCNFET1, NCNFET2, NCNFET3, and PCNFET1. When BL remains at VDD, that is, when BLB remains at GND, the current flows from node Y1 to node Y4, then both ends of memristor M1 are forward voltage and both ends of memristor M2 are reverse voltage, so M1 is in a low resistance state. Logic "1" is written into M1, and M2 is in high resistance state. When BL remains at GND, that is, when BLB remains at VDD, the current flows from node Y4 to node Y1, then both ends of memristor M1 are reverse voltage and both ends of memristor M2 are forward voltage, so M1 is in high resistance state, logic "0" is written into M1, and M2 is in low resistance state at this time.

During the read operation, WL and EN are set to the power supply voltage VDD, while ENB is set to GND to turn on transistors NCNFET1, NCNFET2, NCNFET3, and PCNFET1. A short falling triangular pulse is applied to the BLB, and the BLB discharges to the BL through the memristor and MOS tube. During the reading logic "1", that is, when MEM1 stores "1" and MEM2 stores "0", due to the low resistance state of the former, the voltage drop on MEM1 is much less than that on MEM2, so the voltage on node Y2 is greater than that on Y3. Therefore, it can be obtained that the difference between the voltages on Y2 and Y3 is positive, that is, logic "1". Similarly, when reading logic "0", the voltage difference between Y2 and Y3 is negative, that is, logic "0". It can be seen that the structure improves the reading accuracy to a certain extent by changing the decoding mode of the output voltage, and then makes up for the error caused by the memristor state offset to a certain extent. However, this structure cannot alleviate the problem of resistance state offset of memristor. The problem of resistance offset after multiple readings will lead to the voltage difference gradually approaching 0, which will affect the reading results.

## 4. Design of Stable Continuous Read–Write Circuit Based on Memristor (SCRC)

As mentioned above, the existing read–write circuits still have the problem of resistance state deviation of memristor caused by continuous reading. The above circuits alleviate this problem to a certain extent and do not solve this problem fundamentally.

Figure 6 shows the proposed read–write circuit structure. The main part of the circuit is composed of four memristors (M1, M2, M3, and M4), in which M1 is used as memory to store the signal and M2, M3, and M4 are used as auxiliary components to alleviate the resistance state offset of the storage memristor. M1 and M2 are connected in series with opposite polarity, M3 and M4 are also connected in series with opposite polarity. M1 and M2 are connected in parallel with M3 and M4, and M1 and M3 have opposite polarity, M2 has opposite polarity to M4.

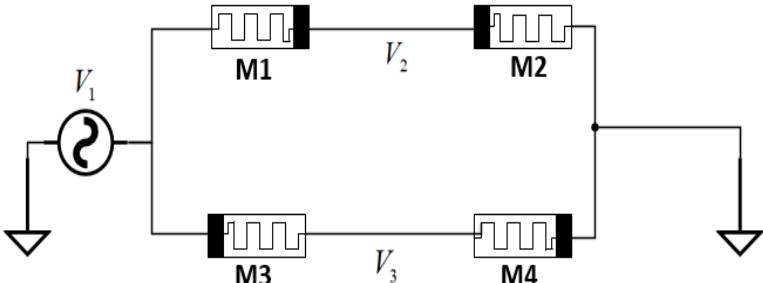

**Figure 6.** Memristor-based read/write circuit with stable successive read operation.

(1) **Write operation**. During the write operation, when the voltage source releases the forward voltage, M1 and M4 are equivalent to applying the positive voltage, the dividing line shifts to the undoped region, the memristors M1 and M4 are in the low resistance state, and the logic "1" is stored in M1. At the same time, M2 and M3 are equivalent to applying the reverse voltage, the dividing line shifts to the doped region, and the memristors M2 and M3 are in the high resistance state. When the voltage source releases a negative voltage, M1 and M4 are equivalent to applying a reverse voltage, the boundary is shifted to the doped region, the memristors M1 and M4 are in a high resistance state, and the logic "0" is stored in M1. At the same time, M2 and M3 are equivalent to applying a positive voltage, the boundary is shifted to the undoped region, and the memristors M2 and M3 are in a low resistance state.

(2) **Read operation.** During the read operation, the input reading signal is a low voltage signal with positive, and the output voltage value $V_o$ is the difference between $V_2$ and $V_3$. When reading logic "1", M1 and M4 are in low resistance state $R_{on}$, M2 and M3 are in high resistance state $R_{off}$, and the resistance values of $V_2$ and $V_3$ are:

$$V_2 = \left( V_1 / \left( R_{on} + R_{off} \right) \right) \cdot R_{off} \tag{5}$$

$$V_3 = \left( V_1 / \left( R_{on} + R_{off} \right) \right) \cdot R_{on} \tag{6}$$

The value of $R_{off}$ is much greater than $R_{on}$, therefore, the value of $V_2$ is close to $V_1$ and the value of $V_3$ is close to zero. Therefore, the output voltage $V_o = V_2 - V_3$ is a positive voltage, which means logic "1".

When reading logic "0", M1 and M4 are in high resistance state $R_{off}$, M2 and M3 are in low resistance state $R_{on}$, and the resistance values of $V_2$ and $V_3$ are:

$$V_2 = \left( V_1 / \left( R_{on} + R_{off} \right) \right) \cdot R_{on} \tag{7}$$

$$V_3 = \left( V_1 / \left( R_{on} + R_{off} \right) \right) \cdot R_{off} \tag{8}$$

The value of $R_{off}$ is much greater than $R_{on}$, therefore, the value of $V_2$ is close to zero and the value of $V_3$ is close to $V_1$. Therefore, the output voltage $V_o = V_2 - V_3$ is a negative voltage, which means logic "0".

Next, we will explain the advantages of this structure through the internal mechanism of memristor. The functional expression of memristor resistance can be obtained from Section 3, as follows:

$$R(w) = \left( R_{on} \cdot w/D + R_{off} \cdot (1 - w/D) \right) \tag{9}$$

where $w$ is the width of the doped region, and the value of $w$ varies with the voltage at both ends of the memristor. When the forward voltage is applied at both ends of the memristor, the value of $w$ increases and gradually approaches to $D$. When the reverse voltage is applied at both ends of the memristor, the value of $w$ decreases and gradually approaches 0. $D$ is the film width of memristor, which is a fixed value. $R_{on}$ is the resistance value of memristor when $w = D$, which is the minimum resistance value. $R_{off}$ is the resistance value of memristor when $w = 0$, which is the maximum resistance value.

When memristor M1 stores logic "1", both M1 and M4 are in low resistance state, assuming that their $w$ values are equal to $w_1$ ($w_1$ is approximately equal to $D$). At this time, both M2 and M3 are in a high resistance state, assuming that their w values are equal to $w_2$ ($w_2$ is approximately equal to 0). At this time, the total resistance of the upper circuit or the lower circuit can be expressed by Equation (10):

$$\begin{aligned} R(w) &= (R_{on} \cdot \tfrac{w_1}{D} + R_{off} \cdot (1 - \tfrac{w_1}{D})) + (R_{on} \cdot \tfrac{w_2}{D} + R_{off} \cdot (1 - \tfrac{w_2}{D})) \\ &= R_{on} \cdot \tfrac{w_1 + w_2}{D} + R_{off} \cdot (2 - \tfrac{w_1 + w_2}{D}) \end{aligned} \tag{10}$$

During the reading operation, the voltage source $V_1$ releases the forward voltage, and the $w$ value of the four memristors will change slightly, and the value of $w_1$ will increase. The increasing offset is defined as $\Delta w_1$. The value of $w_2$ will become smaller, and the decreasing offset is defined as $\Delta w_2$. At this time, the total resistance of the upper circuit or the lower circuit can be expressed by Equation (11):

$$\begin{aligned} R(w) &= (R_{on} \cdot \tfrac{w_1 + \Delta w_1}{D} + R_{off} \cdot (1 - \tfrac{w_1 + \Delta w_1}{D})) + (R_{on} \cdot \tfrac{w_2 - \Delta w_2}{D} + R_{off} \cdot (1 - \tfrac{w_2 - \Delta w_2}{D})) \\ &= R_{on} \cdot \tfrac{w_1 + w_2 + (\Delta w_1 - \Delta w_2)}{D} + R_{off} \cdot (2 - \tfrac{w_1 + w_2 + (\Delta w_1 - \Delta w_2)}{D}) \end{aligned} \tag{11}$$

The voltage expression at $V_2$ is:

$$V_2 = \frac{V_{read}}{R(w)} \cdot R_{M2}(w) = \frac{V_{read}}{R(w)} \cdot (R_{on} \cdot \frac{w_2 - \Delta w_2}{D} + R_{off} \cdot (1 - \frac{w_2 - \Delta w_2}{D})) \tag{12}$$

The voltage expression at $V_3$ is:

$$V_3 = \frac{V_{read}}{R(w)} \cdot R_{M4}(w) = \frac{V_{read}}{R(w)} \cdot (R_{on} \cdot \frac{w_1 + \Delta w_1}{D} + R_{off} \cdot (1 - \frac{w_1 + \Delta w_1}{D})) \tag{13}$$

The expression of output voltage $V_o$ is:

$$\begin{aligned} V_o = V_2 - V_3 &= \tfrac{V_{read}}{R(w)} \cdot (R_{M2}(w) - R_{M4}(w)) \\ &= \tfrac{V_{read}}{R(w)} \cdot (R_{on} \cdot \tfrac{w_2 - w_1 - (\Delta w_2 + \Delta w_1)}{D} + R_{off} \cdot \tfrac{w_1 - w_2 + \Delta w_1 + \Delta w_2}{D}) \\ &= \tfrac{V_{read}}{R(w)} \cdot (R_{off} - R_{on}) \cdot \tfrac{w_1 - w_2 + \Delta w_1 + \Delta w_2}{D} \end{aligned} \tag{14}$$

It can be seen from the above analysis that $w_1$ is approximately equal to $D$ and $w_2$ is approximately equal to 0, therefore $w_1 - w_2 \approx D > 0$, with the condition $R_{off} - R_{on} > 0$. Therefore, it can be obtained from Equation (14) that the output voltage must be greater than 0, which is, the reading result is logic "1", and the offset $\Delta w_1$ and $\Delta w_2$ increase continuously, resulting in the output voltage $V_o$ becoming larger and larger, and the reading result is more accurate. Therefore, our circuit has the positive feedback effect when reading logic "1". When reading logic "0", the output voltage must be less than 0 through the same analysis method above, that is, the reading result is logic "0".

## 5. The Experiment and Results

### 5.1. Data Setup

#### 5.1.1. Memristor Model

LTSpice is used for the modeling and simulation, and the memristor model adopts the bipolar memristor model in literature [35]. Figure 7 shows the LTSpice code of the memristor model, and the simulation results are shown in Figure 2. The memristor model is used for all four circuits in the following simulation experiments. Table 1 shows the detailed parameters of the memristor model. $R_{on}$ = 100 Ω; $R_{off}$ = 16 kΩ; Rinit = 11 kΩ; $D$ = 10 nm; Uv = 10F; $p$ = 1.0, where $R_{on}$ is the resistance value in its low resistance state, $R_{off}$ is the resistance value in the high resistance state, Rinit is the initial resistance value of the memristor, $D$ is the film width of the memristor, Uv is the migration coefficient of the memristor, and p is the parameter of the window function used to model the nonlinear boundary conditions. The window function is:

$$f(x) = 1 - (2x - 1)^{2p} \tag{15}$$

where $x = w/D$, $w$ is the actual width of the doped region ($0 <= w <= D$).

```
.SUBCKT memristor plus minus PARAMS:
+ Ron=100 Roff=16K Rinit=11K D=10N uv=10F p=1.0
Gx 0 x value={I(Emem)*uv*Ron/D**2*f(V(x),p)}
Cx x 0 1 IC={(Roff-Rinit)/(Roff-Ron)}
Raux x 0 1000000
Emem plus aux value={-I(Emem)*V(x)*(Roff-Ron)}
Roff aux minus {Roff}
Eflux flux 0 value={SDT(V(plus,minus))}
Echarge charge 0 value={SDT(I(Emem))}
.func f(x,p)={1-(2*x-1)**(2*p)}
.ENDS memristor
```

**Figure 7.** The model of memristor.

**Table 1.** Memristor parameters setup.

| Variable | Magnitude | Description |
|---|---|---|
| $R_{on}$ | 100 Ω | Min. Resistance of memristor |
| $R_{off}$ | 16 kΩ | Max. Resistance of memristo |
| Rinit | 11 kΩ | The initial resistance value of the memristor |
| $D$ | 10 nm | The film width of the memristor |
| Uv | 10 F | The migration coefficient of the memristor |
| $p$ | 1.0 | The parameter of the window function |

#### 5.1.2. Waveform of Read Signal (Vread)

The input signal (Vread) for read operation is a square wave signal with amplitude of 2 V, period of 20 ms and duty cycle of 50%, which is shown in Figure 8.

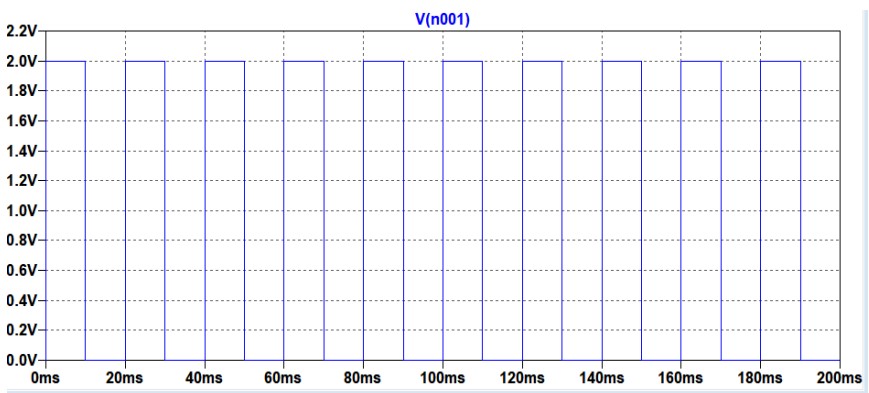

**Figure 8.** Waveform of read signal (Vread).

*5.2. Memristor State Deviation for Continuous Reading Logic "1"*

In order to evaluate the read–write circuit structures, experiments of state deviation for continuous reading logic "1" are performed. Table 2 shows the offset of the output voltage when the logic "1" is read continuously for 100 times. The negative value of the offset represents the voltage attenuation after the continuous read operation, and the positive value represents the voltage enhancement after the continuous reading. It can be seen that NRC and SCRC have the function of positive feedback when reading logic "1". Figure 9 is a waveform diagram of the output voltage when the logic "1" is read continuously for many times. It can be seen from the figure that the initial output voltage of each circuit is different, which is caused by the different partial voltage of different circuit structures. Where (a) and (c) are the output waveform diagrams after reading the logic "1" for NMM and HRC for many times, respectively. It can be seen from the diagram that with the increase of the reading times, that is, the time of inputting the reading signal, the output voltage signal gradually decreases, which will eventually affect the information stored in the memristor. Figure 9b,d are the output voltage waveforms of NRC and SCRC respectively, which shows that NRC and SCRC can enhance the signal "1" stored in the memristor when reading logic "1". This effect is generated by the series connection of memristors with opposite polarity (see Sections 3 and 4 for details).

**Table 2.** Results of continuous reading of logic "1".

|  | Initial Voltage(V) | Voltage after Offset(V) | Offset Time(s) | Offset(V) |
|---|---|---|---|---|
| **NMM** | 0.948 | 0.770 | 6.14 | −0.178 |
| **NRC** | 0.708 | 1.396 | 9.04 | +0.688 |
| **HRC** | 0.951 | 0.774 | 4.96 | −0.177 |
| **SCRC** | 0.939 | 0.983 | 2.82 | +0.044 |

*5.3. Memristor State Deviation for Continuous Reading Logic "0"*

Table 3 shows the offset of output voltage when logic "0" is read continuously until it gradually becomes stable. Figure 10 shows the offset of the resistance state of the memristor when the logic "0" is read continuously for 55 times. The results show that the initial output voltage of each circuit is different, which is caused by the different partial voltage of different circuit structures. Figure 10a is the output waveform diagram after reading the logic "0" for NMM for many times. It can be seen from the figure that with the increase of the reading times, that is, the time of inputting the reading signal, the output voltage signal gradually increases, which will eventually affect the information stored in the memristor. Figure 10b,c show the resistance state offset of NRC and HRC respectively. It can be seen that the offset of NMM and HRC is relatively serious. The offset of NRC is relatively small, but the offset of 0.198 V will still affect the final reading result. Figure 10d shows the output

voltage at $V_2$ after the continuous reading logic "0" operation of SCRC. According to the analysis in Section 4, the voltage at $V_2$ when reading "0" is the same as that at $V_3$ when reading 1, so the output waveform at $V_3$ when reading "0" is the same as that in Figure 9d. From the data in Tables 2 and 3, it can be seen that the voltage offset of SCRC when reading "0" is 0.122 V−0.044 V = 0.078 V, which has greater stability.

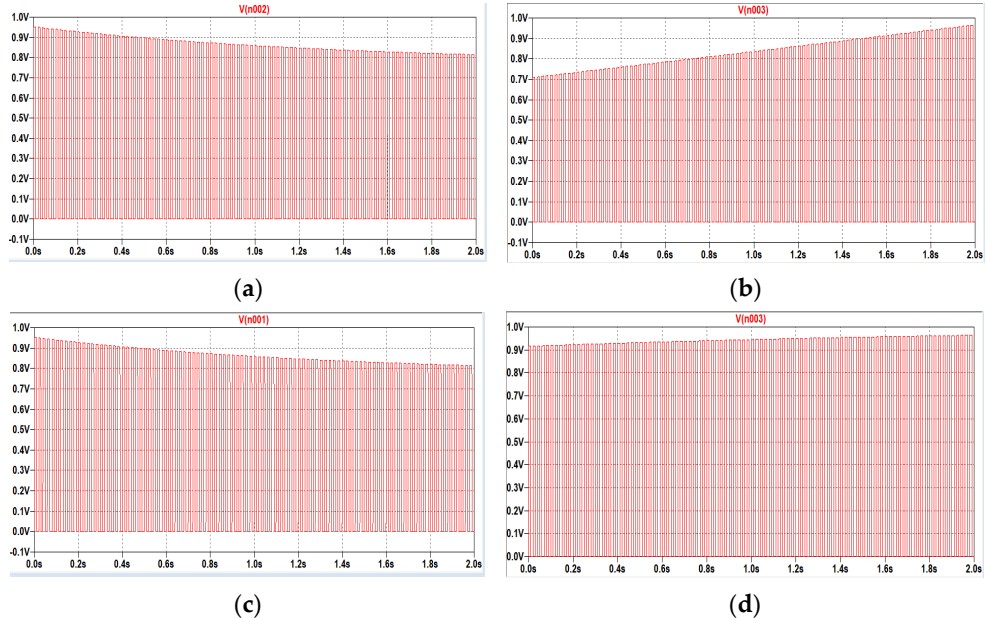

**Figure 9.** (**a**) Output waveform diagram of NMM after reading logic "1"; (**b**) output waveform diagram of NRC after reading logic "1"; (**c**) output waveform diagram of HRC after reading logic "1"; and (**d**) voltage waveform diagram at $V_2$ of SCRC after reading logic "1".

**Table 3.** Results of continuous reading of logic "0".

|  | Initial Voltage(mV) | Voltage after Offset(mV) | Offset Time(s) | Offset(V) |
|---|---|---|---|---|
| **NMM** | 18 | 573.53 | 1.143 | 0.556 |
| **NRC** | 223.18 | 421.65 | 16.36 | 0.198 |
| **HRC** | 59.47 | 414.4 | 2.935 | 0.355 |
| **SCRC** | $V_2$ = 253.82 $V_3$ = 939.17 | $V_2$ = 376.23 $V_3$ = 983.32 | 14.92 | $\Delta V_2 - \Delta V_3$ = 0.078 |

### 5.4. Power Consumption and Performance

It can be seen from Table 4 that the power consumption of the write operation of NMM and NRC is the same, because the structure of the write circuits of both circuits is exactly the same. The power consumption of the write logic "1" is higher than that of the write logic "0", while the power consumption of the write logic "0" and logic "1" of HRC and SCRC is the same. The reason is that both of them are connected in parallel with memristors. When writing logic "0" and logic "1", the total resistance of the memristor is the same, and the imported voltage is the same, so the power consumption is the same. When reading, the power consumption of NRC is the smallest. HRC and SCRC reduce the impact of state offset with the overhead of a small power consumption, though the power consumption of the improved circuit (SCRC) is reduced to a certain extent. In terms of read–write delay, due to the write circuits of NMM and NRC are the same, the delay is the same. The delay of HRC and SCRC is almost the same during the read–write operation. There are more MOS transistors in the structure of HRC, therefore, the delay is relatively high.

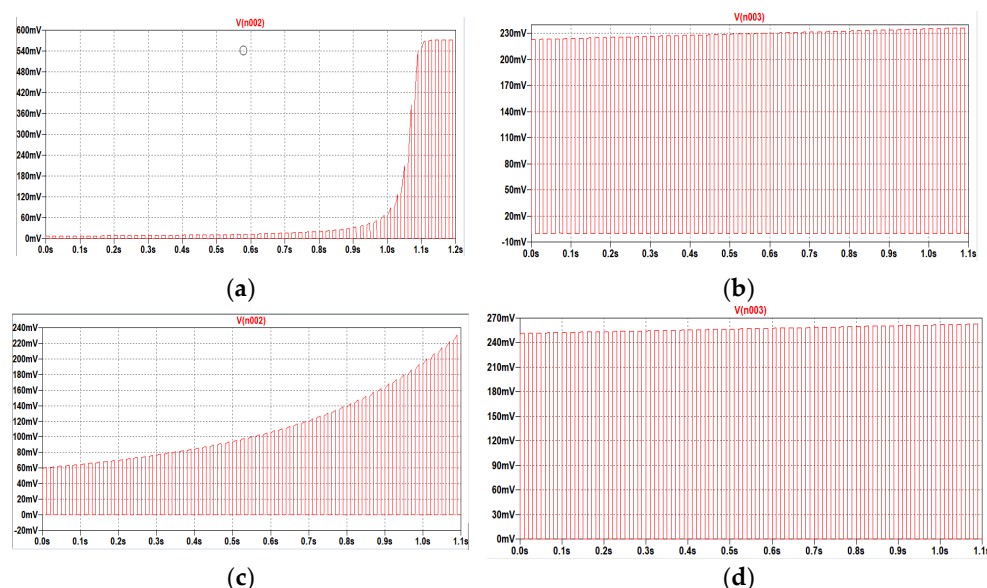

**Figure 10.** (**a**) Output waveform diagram of NMM for reading logic "0"; (**b**) output waveform diagram of NRC for reading logic "0"; (**c**) output waveform diagram of HRC for reading logic "0"; and (**d**) voltage waveform diagram at $V_2$ of SCRC for reading logic "0".

**Table 4.** Comparison of power consumption and delay of read–write operation in different circuits.

|  | **NMM** | **NRC** | **HRC** | **SCRC** |
|---|---|---|---|---|
| **Power for writing "1"(uW)** | 125 | 125 | 31.53 | 23.57 |
| **Power for writing "0"(uW)** | 25.37 | 25.37 | 31.36 | 23.24 |
| **Power for reading "0"(uW)** | 4.45 | 0.28 | 4.25 | 3.68 |
| **Power for reading "1"(uW)** | 20.31 | 0.45 | 4.18 | 3.56 |
| **Delay time for writing "0"(ns)** | 0.13 | 0.13 | 1.43 | 1.35 |
| **Delay time for writing "1"(ns)** | 0.39 | 0.39 | 1.57 | 1.31 |
| **Delay time for reading "0"(ns)** | 3.58 | 0.046 | 0.82 | 0.63 |
| **Delay time for reading "1"(ns)** | 3.16 | 0.014 | 0.75 | 0.58 |

*5.5. Design Exploration*

We set different $R_{on}$ and $R_{off}$ resistance values to simulate the influence of $R_{on}$ and $R_{off}$ resistance on power consumption. The results (Table 5) show that the greater the resistance value of the memristor, the smaller the power consumption of the circuit. Therefore, in practical application, the resistance value of the memristor can be appropriately increased to reduce the power consumption of the resistance, but when the resistance value is too large, the current in the circuit will be very small, which is not conducive to the monitoring of the output terminal. Therefore, it is necessary to balance the relationship between them in practical application in order to achieve the best effect. Since the large structure of the circuit is a parallel circuit, the total voltage of the upper circuit and the lower circuit is the same, which will make the resistance states of M1 and M4 the same, and the resistance states of M2 and M3 the same, i.e., Rm1 = Rm4, Rm2 = Rm3. Therefore, the power consumption when writing logic "1" and logic "0" is the same, as well as when reading.

**Table 5.** Reading and writing power consumption at different resistance values.

|  | $R_{on}$ = 10 Ω $R_{off}$ = 1.6 kΩ | $R_{on}$ = 100 Ω $R_{off}$ = 1.6 kΩ | $R_{on}$ = 10 Ω $R_{off}$ = 16 kΩ | $R_{on}$ = 100 Ω $R_{off}$ = 16 kΩ |
|---|---|---|---|---|
| **Power for writing "1" (uW)** | 194.62 | 173.58 | 25.39 | 23.24 |
| **Power for writing "0" (uW)** | 194.83 | 174.21 | 25.67 | 23.57 |
| **Power for reading "1" (uW)** | 27.23 | 24.63 | 4.15 | 3.68 |
| **Power for reading "0" (uW)** | 27.58 | 24.81 | 4.23 | 3.56 |

## 6. Conclusions

We are committed to reducing the memristor state shift caused by continuous reading operations, in order to improve the accuracy of read and write. Our major achievement is the proposed read–write circuit model based on the parallel structure of memristors with opposite polarity. The logic "1" and logic "0" of the structure are, respectively, represented by the positive and negative of the voltage difference between the upper and lower circuits in the parallel structure, and each circuit is connected in series by two memristors with opposite polarity. The two memristors with opposite polarity can compensate each other for the loss caused by the resistance state shift, so the fault tolerance of the circuit can be effectively improved. Compared with the circuit proposed in reference [18,21,27], our circuit is more stable during continuous reading and writing. The experimental results show that after continuous reading, the offset of our structure output voltage is reduced to 78 mv. The impact of this level of offset on the final reading result is almost negligible, and it is significantly lower than the lowest offset (198 mv) of the three circuits in [18,21,27]. In terms of reading delay and power consumption, our circuit is significantly better than those in [18,27]. Although the power consumption and read–write delay of the proposed circuit are slightly higher than those of the circuit in [21], this is a concession to reduce the resistance state offset. In practical applications, these indicators should be balanced according to the real situation. In the following research, we are dedicated to optimizing the read–write delay and power consumption as much as possible without increasing the resistance state offset.

**Author Contributions:** Conceptualization, W.L. and N.B.; methodology, W.L.; software, N.B.; validation, W.L. and N.B.; formal analysis, W.L.; investigation, T.Z., Y.S. and X.Z.; data curation, N.B., T.Z., Y.S. and X.Z.; writing—original draft preparation, N.B.; writing—review and editing, W.L.; supervision, W.L. All authors have read and agreed to the published version of the manuscript.

**Funding:** This research received no external funding.

**Data Availability Statement:** The memristor model and related parameters used in this paper can be referred to [35]. The structures and related parameters of the three circuits used for comparative evaluation can be referred to [18,21,27] Other data will be used for future research and cannot be shared temporarily.

**Conflicts of Interest:** The authors declare no conflict of interest.

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
