# Peer review of "Memristor-Based Read/Write Circuit with Stable Continuous Read Operation"

_electronics, doi:10.3390/electronics11132018_

Round 1

Reviewer 1 Report

Full Title: OPP: Memristor-Based Read/Write Circuit with Stable Continuous Read Operation

In the considered paper, the authors introduced and investigated a new model of memristor.

In general, the paper is too confusing and the results are not well presented.

  • The use of abbreviations in the title is not suitable.
  • What equation is able to obtain Fig.2:
  • In the experimental part (section 4), the unit used for resistors is not correct.
  • Provide a full description of the simulation environment where the experimental part has been addressed.
  • Provide a table where your memristor model as well as your result will be compared with some similar work on memristor design.
  • In some sections, the word "chapter" appears and should be removed.
  • The writing should be further improved since there are some typographical errors.

Author Response

Thank you very much for your suggestions. Your suggestion has provided great help for the revision of our paper Please see the attachment for the reply to your comments.

Reviewer 2 Report

Please find attached my comments.

Author Response

(The authors gave the same response as above.)

Reviewer 3 Report

The paper describes the structure based on the use of memristive elements in the realization of memory elements. The authors in the text use abbreviations, and even in the title without defining them. In most part, the authors are dealing so far by a well-known solution, while contributions from the author is not supported by the theory that confirms the correctness of the proposed approach. Simulation checks confirm the maintenance of voltage level during reading and writing operations, but the structure is very complex and is unclear how it would be realized in practice. The results shown in tables shows additional delay and increasing consumption in relation to some known solutions. All of this does not justify in a clear and precise way the concept that is proposed.

Author Response

Thank you very much for your suggestions. Your suggestion has provided great help for the revision of our paper. Please see the attachment for the reply to your comments.

Round 2

Reviewer 1 Report

paper can be accepted in the present form

Author Response

Thank you very much for your affirmation of our work. Your comments have provided great help to the revision of our paper.We wish you good health and all the best!

Reviewer 3 Report

The work has been significantly improved compared to the previous version. Simple Present Tense should be used when you discuss your circuit. Some grammatical errors should be corrected. It is necessary to use indices in several places in the work when marking voltages and currents in the circuit, and not to write numerous values in a line with the mark.

Author Response

Thank you very much for your comments. We have corrected the tense and grammatical errors in the paper. The problem you mentioned about marking is indeed our negligence and has been modified. See the revised paper for details. Your comments have provided great help to the revision of our paper. Thanks!

We wish you good health and all the best!